# Deep Reinforcement Learning for Online Control of Stochastic Partial Differential Equations

**Erfan Pirmorad**[1],[*] **Faraz Khoshbakhtian**[1], **Farnam Mansouri**[2,3], **Amir-massoud Farahmand**[3, 2, 1]
[1]Department of Mechanical & Industrial Engineering, University of Toronto
[2]Department of Computer Science, University of Toronto
[3]Vector Institute
{erfan.pirmorad, faraz.khoshbakhtian, farnam.mansouri}@mail.utoronto.ca,
farahmand@vectorinstitute.ai

## Abstract

In many areas, such as the physical sciences, life sciences, and finance, control approaches are used to achieve a desired goal in complex dynamical systems governed by differential equations. In this work we formulate the problem of controlling stochastic partial differential equations (SPDE) as a reinforcement learning problem. We present a learning-based, distributed control approach for online control of a system of SPDEs with high dimensional state-action space using deep deterministic policy gradient method. We tested the performance of our method on the problem of controlling the stochastic Burgers' equation, describing a turbulent fluid flow in an infinitely large domain.

## 1 Introduction

Partial differential equations (PDE) are commonly used to explain complex dynamical systems such as fluid dynamics, electromagnetism, etc. While PDEs successfully describe a wide range of dynamical systems, random perturbations require further modelling in the governing equations of some dynamical systems, e.g., turbulent flows. Stochastic partial differential equations (SPDEs) are used to model the inherent stochasticities in dynamical systems (Walsh [1986]). Drawing by recent advances in design and control of the dynamical systems in engineering applications, PDE control has attracted a lot of attention (Krstić [2008], Ahuja et al. [2011], Brunton and Noack [2015], Burns and Hu [2013]); whereas control of SPDEs remains scarce. PDE control methods are incorporated successfully in many engineering applications; however, the majority of them require far too much knowledge of the system, including a complete model of the PDE and the design of a customized controller, both of which are computationally expensive and not robust to environmental changes.

Reinforcement learning (RL) was recently used for PDE control with the application of heating, ventilating, air conditioning (HVAC) control design in a room as an example(Farahmand et al. [2016, 2017], Pan et al. [2018]). In these works, the heat transfer equation with a pre-determined flow field velocity was formulated as an MDP, and RL algorithms used to control the state of the ventilator. Unlike PDE control, there have been few studies on controlling SPDEs. Due to the nonlinearity and randomness of the system, such a problem is extremely challenging for traditional control methods to address (Rosseel and Wells [2012], Øksendal [2005]). In this work, we present an RL framework for online control of stochastic dynamical systems. Inspired by the promising performance of recent RL algorithms in PDE control, we formulate the problem of controlling a SPDE as an MDP with an infinite dimensional state-action space, and use Deep Deterministic Policy Gradient (DDPG) (Lillicrap et al. [2016]) to train our agent. We evaluate our framework to control the stochastic Burgers' equation, by damping the shock wave induced by inherent perturbations.

---

[*]First three authors have equal contribution

DLDE Workshop in the 35th Conference on Neural Information Processing Systems (NeurIPS 2021).

## 2 Methodology

We consider a (controlled) stochastic dynamical system governed by

$$\frac{\partial \mathbf{u}(\mathbf{x},t)}{\partial t} = \mathcal{F}\left(\mathbf{x}, \mathbf{u}, \frac{\partial \mathbf{u}}{\partial \mathbf{x}}, \left(\frac{\partial^2 \mathbf{u}}{\partial x_i \partial x_j}\right)_{i,j=1:n}, \cdots; \mathbf{p}\right) + \xi(\mathbf{x},t) + \mathbf{f}(\mathbf{x},t), \tag{1}$$

where $\mathbf{u}(\mathbf{x},t)$ denotes the state variable of the dynamical system which depends on spatial free variables $\mathbf{x} = (x_1, \cdots, x_n)$, within an $n$-dimensional domain $\Omega$, $x_{1:n} \in \Omega \subset \mathcal{R}^n$, and time $t \in \mathcal{R}^+$, and its derivatives with respect to the free variables. $\mathcal{F}(.)$ is a nonlinear general function of the derivatives parameterized by $\mathbf{p}$. In the above equation $\xi(\mathbf{x},t)$ is a multidimensional noise that is white in time and correlated in space, and $\mathbf{f}(\mathbf{x},t)$ is a control input forcing function, which will be further discussed later.

The above SPDE is solved, either numerically or analytically, on domain $\Omega$ bounded to the boundary $\partial\Omega = \Gamma$, with boundary condition $u_\Gamma$. An example of the control problem here can be defined as setting the state variable to a desirable value $\mathbf{u} = \mathbf{u}^*$, within an arbitrary controlled sub-domain of $\Omega_c$ bounded to the boundary $\partial\Omega_c = \Gamma_c$, at all time $t$. A realistic example of this sort of control problem is an HVAC unit system where we attempt to set the temperature to a comfortable value, in a specific part of a building, in the presence of an air flow. The desired control problem can be formally defined as the following optimal control problem:

$$\inf_{\mathbf{f}} \; J(\mathbf{u}, \mathbf{u}^*) \triangleq \frac{1}{2} \int \int_{\Omega_c} (\mathbf{u}(\mathbf{x},t) - \mathbf{u}^*(\mathbf{x},t))^2 \, d\Omega \, dt$$

$$\text{s.t.} \;\; \frac{\partial \mathbf{u}(\mathbf{x},t)}{\partial t} = \mathcal{F}\left(\mathbf{x}, \mathbf{u}, \frac{\partial \mathbf{u}}{\partial \mathbf{x}}, \left(\frac{\partial^2 \mathbf{u}}{\partial x_i \partial x_j}\right)_{i,j=1:n}, \cdots; \mathbf{p}\right) + \xi(\mathbf{x},t) + \mathbf{f}(\mathbf{x},t). \tag{2}$$

To solve this (optimal) control problem, we propose using an RL approach. To this end, the continuous space-time SPDE is discretized in both space and time by a numerical solver, and the discretized SPDE is formulated as a discrete space-time, continuous state-action space, discounted MDP $(\mathcal{S}, \mathcal{A}, \mathcal{P}, \mathcal{R}, \gamma)$ (Sutton and Barto [2018], Szepesvári [2010]), where state space $\mathcal{S}$ and action space $\mathcal{A}$ are vector functions depending on the function space of SPDE $\mathcal{F}(\mathcal{Z})$, defined over the domain $\mathcal{Z}$ of SPDE. The current value of the target variable, within $\Omega_c$ at each discrete time $t$, is the current state $s_t = \mathbf{u}(\mathbf{x},t)$. Assuming that the system is equipped with a number of actuators, a localized control input function is the continuous action $a_t = \mathbf{f}(\mathbf{x},t)$, within the controlled domain $\Omega_c$. A scalar reward function is defined to penalize the RL agent when the target variable deviates from the desired value, given by:

$$r_t = -\left[\frac{1}{2}\int_{\Omega_c}(\mathbf{u}(\mathbf{x},t) - \mathbf{u}^*(\mathbf{x},t))^2 \, d\Omega + \frac{1}{2}\lambda \|\mathbf{f}(\mathbf{x},t)\|^2\right], \tag{3}$$

which is negative of the cost function, regularized by the cost of choosing an action (which is small or zero for cheap controls and large for expensive controls). A new state given the chosen action and the previous state is sampled as $s_{t+1} \sim \mathcal{P}(\cdot \mid s_t, a_t)$, where the transition probability kernel $\mathcal{P}$ depends on the SPDE.

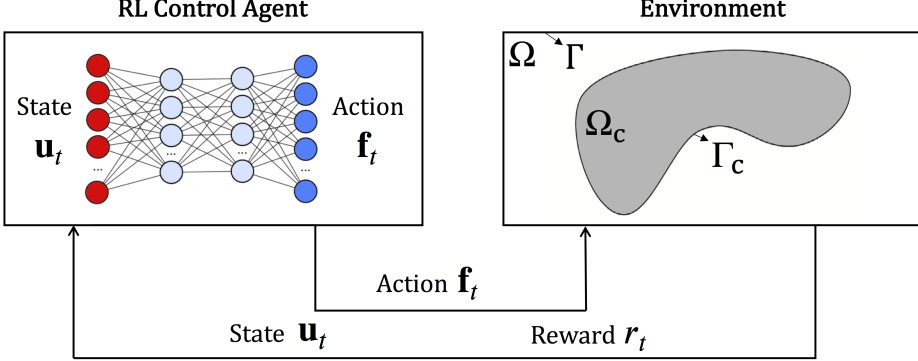

Figure 1: Schematic of the RL control agent interacting with an environment that is a 2D domain $\Omega$ governed by an SPDE, and the control sub-domain $\Omega_c$, where a desired condition is enforced.

## 3 Experiments

The potential benefits of SPDE control is known to be significant in controlling turbulent flows that occur in various engineering applications (Naseri and Malek [2014]). Attempts to control turbulent flows in engineering applications have focused on the manipulation of coherent structures, such as a shock wave. A shock wave is a region within the fluid flow where physical conditions undergo an abrupt change because of high flow variable gradients, and this damages structures as it propagates. 1D Stochastic Burgers' equation (SBE) is a simple model that best describes this phenomenon:

$$\frac{\partial u}{\partial t} + \frac{\partial}{\partial x}\frac{u^2}{2} = \nu\frac{\partial^2 u}{\partial x^2} + \epsilon\eta_t(x), \quad \eta_t = \frac{\partial\dot{W}}{\partial x}, \quad W \sim \mathcal{N}\left(0, \sigma^2\right), \tag{4}$$

where $\eta_t(x)$ is a space-time white noise with the intensity of $\epsilon$, and it is realized as the generalized derivative of the Brownian sheet, i.e. $\eta_t(x) = \partial_t\partial_x W_t(x)$, where $W_t(x)$ is a Gaussian process. Due to the presence of nonlinear convective terms, solution of the 1D Burgers' equation is prone to exhibit a chaotic behaviour. Here, we test our proposed RL framework to control the fluid flow governed by the SBE to damp the developed shock wave and stabilize the system in an online manner, and compare it against previous non-RL approaches (Choi et al. [1993], Munteanu [2019]). A distributed control method is used by introducing an external forcing term, $f(x, t)$, as a control input of the system. The control problem here is formalized to find the proper $f(x, t)$ applied to the fluid flow in the solution domain to squash the shock wave at each time. Squashing in our terminology means $\inf_f \mathbb{E}[\int_{\Omega_c} (u(x, t) - \bar{u})^2 \, dx]$ in each time step, where $\bar{u} = \mathbb{E}[\frac{1}{2\pi}\int_{\Omega_c} u(x, t)dx]$. The described control problem above can be expressed as the following inverse problem:

$$\inf_f \; J(u, f) \triangleq \int \left[\frac{1}{2}\int_{\Omega_c} (u(x, t) - \bar{u})^2 \, dx + \frac{1}{2}\lambda\left\|f\right\|^2\right] \, dt \quad \text{on } \Omega_c$$

$$\text{s.t.} \begin{cases} \frac{\partial u}{\partial t} + \frac{\partial}{\partial x}\frac{u^2}{2} = \nu\frac{\partial^2 u}{\partial x^2} + \epsilon\eta_t(x) + f(x, t), & 0 < x < 2\pi \\ u(x, t = 0) = u_0(x) \\ u(x = 0, t) = u(x = 2\pi, t), & \text{Periodic Boundary condition} \end{cases} \tag{5}$$

### 3.1 Training Procedure

**SPDE solver:** A uniform computational grid of $n_x = 151$ points is used for spatial discretization. A Crank-Nicolson method (Crank et al. [1947]) in time and second-order centred differences in space are used to discretize the equation, an explicit iterative method is used to solve the discretized nonlinear equation. Instantaneous velocity field for evolution of a shock wave is shown in Figure 2. It is observed that the shock wave energy dissipates over time by the structural diffusivity of system.

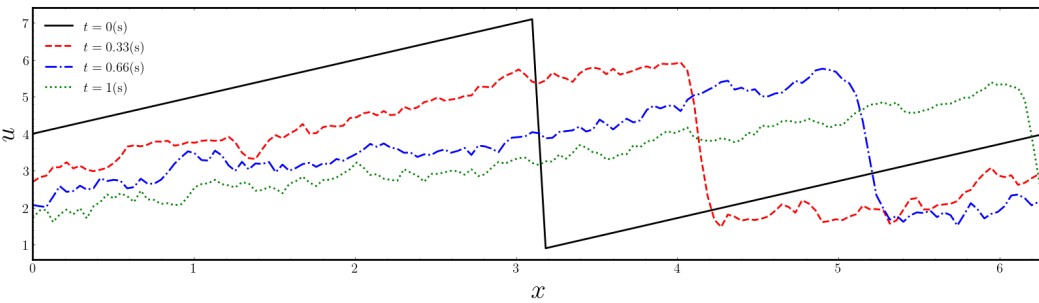

Figure 2: Instantaneous velocity field for evolution of a shock wave in space and time in four different times.

**RL:** Given the continuous state-action nature of the SPDE, we use a deep deterministic policy gradient (DDPG) approach to train the RL agent. We implemented an actor/critic DDPG framework according to Lillicrap et al. [2016]. The DDPG states and actions are $u(.; t)$ and $f(.; t)$, respectively. To facilitate this algorithm implementation in the real world, the action space is limited to a set of 4-interval piecewise constant functions. Similar fully connected 2-layer networks are used as the

actor and critic networks architecture, that consist of hidden layers of ReLU units, and a linear output layer. Network parameters are initialized using Xavier initialization (Glorot and Bengio [2010]), and Adam optimizer(Kingma and Ba [2015]) is used for optimization. An experience replay with the buffer size of 1000000 is used. Details on the hyperparameters used for training is provided in Appendix A.

## 3.2   Results

Here we show some preliminary results on controlling the SBE, and compare them against benchmarks. Figure 3(a) and Figure 3(b) show the free evolution of the shock wave (black line), evolution under online control of the RL agent (red dotted-line), and the input control function at $t = 0.8$ and averaged over $N = 20$ runs with $90\%$ confidence interval, respectively. It can be seen that the control input function damps the shock wave and reduces the gradients throughout the domain. We compare the average return for $N_{eps} = 100$ episodes obtained by the RL against uncontrolled system, and the suboptimal control approach used by Choi et al. [1993], given the same form of piecewise constant functions as the action space in Figure 3(c). Finally, the average return for three gradually more expressive action functions are compared in Figure 3(d).

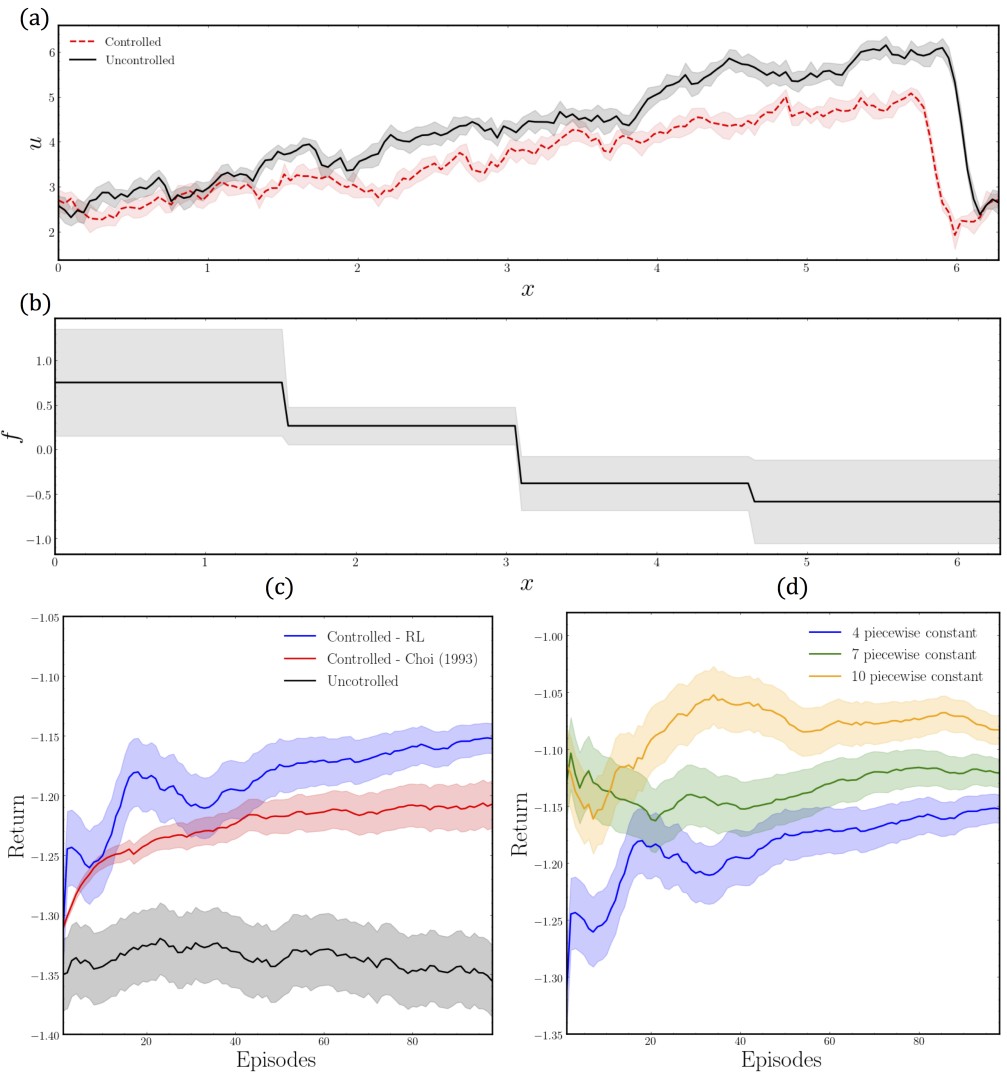

Figure 3: (a) Velocity fields $u(.; t = 0.8)$ (state) for uncontrolled (black) and controlled (red) evolution of the shock wave described by the SBE. (b) Calculated control input $f(.; t = 0.8)$. Average return for (c) the RL agent (blue), the controller of Choi et al. [1993] (red), and the uncontrolled environment (black); and (d) different form of action functions with 4, 7, and 10 dimensional actions space.

# 4 Conclusion

In this work, we presented a learning-based approach, based on reinforcement learning, to control behaviour of a SPDE that describes a noisy dynamical system. Preliminary results for online control of the 1D SBE showed promising results in damping the shock wave and reducing sharp gradients. Although boundary control approaches are more practical in fluid mechanics, we adopted a distributed control approach for the case of 1D SBE because it is defined in an infinitely large domain, represented by periodic boundary conditions on a $2\pi$ length cell. Our proposed RL control approach is easily extendable to a boundary control strategy to control bounded systems. We will extend this work to more realistic higher dimensional SPDEs in our future work.

# 5 Acknowledgment

AMF acknowledges the funding from the Canada CIFAR AI Chairs program.

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

# A   Hyperparameters

Table 1: Parameter values for 1D SBE equation training

| Parameter | Value | Description |
|-----------|-------|-------------|
| $t_{start}$ | 0 | Initial time |
| $t_{end}$ | 2 | Finishing time |
| $\delta_t$ | 2 | Episode length |
| $n_x$ | 151 | Number of spatial grids |
| $x$ | $[0, 2\pi]$ | Lower and upper bounds of $x$ |
| $u$ | [-8, 8] | Lower and upper bounds of $u$ |
| $f$ | [-10,10] | Lower and upper bounds of $f$ |
| $\nu$ | 0.01, 1, 10 | Diffusion coefficient |
| $\epsilon$ | 0.01 | Magnitude of the Gaussian noise |
| $\lambda$ | 0.2 | Regularizer weight |
| $\alpha$ | 2.5e-5 | Actor network learning rate |
| $\beta$ | 2.5e-4 | Critic network learning rate |
| $layer1 - size$ | 400 | First layer dimension for both Actor/Critic |
| $layer2 - size$ | 300 | Second layer dimension for both Actor/Critic |
| $\tau$ | 0.1 | Target network update parameter |

