# OpenReview forum: "Deep Reinforcement Learning for Online Control of Stochastic Partial Differential Equations"
_NeurIPS.cc/2021/Workshop/DLDE — DLDE Workshop -- NeurIPS 2021 Spotlight_

### Official Review · Reviewer_pHBP · 2021-09-29
**This paper explores control for Stochastic PDEs using Reinforcement Learning. The paper reads well and shows promising results.**

**Confidence:** 3

**Review:**

This paper uses Reinforcement Learning to control Stochastic Partial Differential Equations, extending previous work on PDEs to include the stochastic element. The authors test their model on the one-dimensional Stochastic Burger's equation, showing promising results. The paper is well written and describes the methodology well.


I have a couple of points I'd like to clarify and some typos:

- You've said in line 38 that $\mathcal{F}(.)$ is a differential operator. I just want to check that is the case because in equation 1, $\mathcal{F}$ takes the partial derivatives as input, does that not simply make it a general function of the derivatives, rather than an operator?

- In equation 2, I believe you need to have "s.t. $\frac{\partial u}{\partial t} = \mathcal{F} (x_i...$" rather than "s.t. $\mathcal{F}(x_i ...$"

- In equation 3, you have the reward being a function of $\mathbf{x}$ and $t$ shouldn't it just be a function of $t$, given that you are integrating over space?

- Line 77 needs a period between $\bar{u} = \mathbb{E}[\frac{1}{L}\int_{\Omega}udx]$ and "The"

- Line 85 should say "state-action" rather than "sate-action".

This paper will be great for the workshop and I would be interested to see future iterations of the work.


**Score:**

4: Very good paper

---

### Official Review · Reviewer_823q · 2021-10-03
**Deep RL to solve Stochastic PDE control.**

**Confidence:** 4

**Review:**

### Summary

Deep RL solution for a Stochastic PDE control problem.

### Comments

**pros**: Benchmarked against previous non DRL system, over 20 seeds.

**cons**: No major criticism.

**Score:**

4: Very good paper

---

### Official Review · Reviewer_gfYu · 2021-10-11

**Confidence:** 3

**Review:**

The authors propose to use deep reinforcement learning for controlling stochastic partial differential equations. They use deep deterministic policy gradient to control the stochastic Burger’s equation. They compare their method with an optimal control approach used by Choi et al. [1993].

Comments:
-	In the Abstract, the Conclusion and in line 73, the authors indicate that they present a distributed control approach. I would suggest adding a paragraph justifying why the controller is distributed.
-	Since the authors’ proposed method is compared with the controller of Choi et al. [1993], it might be useful to add a description of the latter (maybe as part of the Appendix). Moreover, the velocity field obtained when using Choi’s controller can be added to Fig. 3(a).
-	In line 88-89, the authors indicate that, to facilitate this algorithm implementation in the real world, they limit the action space. However, how is the forcing term applied in a real world scenario? What actuators could produce such a piecewise $f(.;.)$?
-	In the literature review, the authors did not cover any control theoretic method for similar systems. I would suggest including some of them. This would also improve the motivation of using RL over classic control methods.

Other (minor) comments:
-	In equation 1, the notation ${\bf f}(x_i, t)_{i=1:n}$ is not clear.
-	In equation 1, $\bf f$ is a function of two scalar values. In line 56, $\bf f$ is a function of a vector in $\mathrm{R}^n$ and a scalar value. The notation should be consistent.
-	In Table 1, it is not clear why there are three different diffusion coefficients. Moreover, it is indicated the maximum bounds of $x$, $u$ and $f$, but not the minimum bounds. Finally, I would suggest indicating the number of layers of the networks used; even if from the table, a reader may guess that a 2-layer network was used.
-	In line 77, $L$ is not defined.
-	In line 80, it should be points instead of point.


**Score:**

2: Borderline paper

---

### Official Review · Reviewer_T8XC · 2021-10-11
**Deep Deterministic Policy Gradient (DDPG) RL is proposed to address the problem of control of Stochastic PDEs. The proposed framework is applied to the 1D-Burges equation, showing a potentially promising foundation for future applications.**

**Confidence:** 3

**Review:**

Pros: Using DDPG to optimize control of stochastic systems is potentially a promising approach. The formulation proposed is interesting and warrants further research.

Cons: More details on the implementation are needed. An evaluation of the limitations of DDPG for control of SPDEs is missing. A comparison with other approaches is needed. For example:

Variational Optimization Based Reinforcement Learning for Infinite Dimensional Stochastic Systems. Ethan N. Evans, Marcus A. Periera, George I. Boutselis, Evangelos A. Theodorou Proceedings of the Conference on Robot Learning, PMLR 100:1231-1246, 2020.

Why a buffer size of 1000000? How sensitive are the experimental results to hyperparameter selection?

This paper could be an interesting contribution to the workshop. I look forward to learning more details about the proposed approach.


**Score:**

3: Good paper

---

### Decision · Program_Chairs · 2021-10-15

**Decision:**

Accept (Spotlight)

**Comment:**

Reviews were generally very positive. The main concerns were the low-dimensionality of the example problem and the need for better comparison to alternative DL and non-DL methods for this application. Since most reviewers were eager to hear more, this submission will make a good spotlight for the workshop. The authors may wish to address the concerns raised in the reviews to maximize the impact of their presentation.